# Transformation of doped graphite into cluster-encapsulated fullerene cages

Marc Mulet-Gas[1], Laura Abella[2], Maira R. Cerón [3], Edison Castro [3], Alan G. Marshall[1,4], Antonio Rodríguez-Fortea [2], Luis Echegoyen[3], Josep M. Poblet [2] & Paul W. Dunk [1]

An ultimate goal in carbon nanoscience is to decipher formation mechanisms of highly ordered systems. Here, we disclose chemical processes that result in formation of high-symmetry clusterfullerenes, which attract interest for use in applications that span biomedicine to molecular electronics. The conversion of doped graphite into a $C_{80}$ cage is shown to occur through bottom-up self-assembly reactions. Unlike conventional forms of fullerene, the iconic Buckminsterfullerene cage, $I_h$-$C_{60}$, is entirely avoided in the bottom-up formation mechanism to afford synthesis of group 3-based metallic nitride clusterfullerenes. The effects of structural motifs and cluster–cage interactions on formation of compounds in the solvent-extractable $C_{70}$–$C_{100}$ region are determined by in situ studies of defined clusterfullerenes under typical synthetic conditions. This work establishes the molecular origin and mechanism that underlie formation of unique carbon cage materials, which may be used as a benchmark to guide future nanocarbon explorations.

[1] National High Magnetic Field Laboratory, Florida State University, Tallahassee, FL 32310, USA. [2] Departament de Química Física i Inorgànica, Universitat Rovira i Virgili, Tarragona 43007, Spain. [3] Department of Chemistry, University of Texas at El Paso, El Paso, TX 79968, USA. [4] Department of Chemistry and Biochemistry, Florida State University, Tallahassee, FL 32306, USA. Marc Mulet-Gas and Laura Abella contributed equally to this work. Correspondence and requests for materials should be addressed to L.E. (email: echegoyen@utep.edu) or to J.M.P. (email: josepmaria.poblet@urv.cat) or to P.W.D. (email: dunk@magnet.fsu.edu)

Fullerenes that encapsulate clusters of atoms represent a fundamental interest in chemistry, materials, and carbon science due to their unique properties and nanoscale structures[1–3]. Compounds that entrap the trimetallic nitride cluster are among the most intensively studied form of molecular nanocarbon because they offer promise as contrast agents and other biomedical diagnostics, photovoltaics, and molecular electronics[4–7]. In particular, $Sc_3N@I_h$-$C_{80}$ mysteriously forms as the "third most abundant fullerene", only empty $C_{60}$ and $C_{70}$ have been isolated in higher yield[8, 9]. The $M_3N$ (M = metal) cluster imparts stability to cage sizes from ~$C_{70}$ to $C_{100}$ and donates six electrons to the carbon cage[10, 11]. Nitride clusterfullerene (NCF) compounds possess diverse structural motifs that are relevant to other carbon networks, such as nanotubes and graphene. For example, non-isolated pentagon rule (non-IPR), $Sc_3N@D_3$-$C_{68}$, as well as an isomer of the $C_{66}$ cage exhibit multiple configurations of fused pentagons[12–14]. Very recently, a heptagon-containing structure was characterized as the NCF, $Sc_2LaN@C_{80}(hept)$[15].

An understanding of how these compounds form by simple vaporization of doped graphite is paramount because the intrinsic mechanisms and chemical principles that control formation may be exploited to create entirely new forms of nanomaterials and overcome obstructions in synthesis of cluster-encapsulated carbon materials. Recently, top-down proposals have been rationally inferred as a possible route to formation of the archetypal NCF, $M_3N@I_h$-$C_{80}$, and other high-symmetry fullerenes based on molecular evidence, computational studies, and observations of graphene under electron beam irradiation[16–19]. In this case, carbon sheets are envisioned to warp into giant cages and subsequently shrink into the icosahedral $C_{80}$ cage. However, it remains unknown how an icosahedral cage that entraps a four-atom cluster may self-assemble from graphite vaporization, because in situ studies are not possible by conventional synthesis methods. Such in situ studies are a challenging endeavor, but are crucial by virtue that carbon is an extraordinary element that possesses the tendency to exhibit different chemistry under conditions that diverge from characteristic synthesis due to its versatile bonding properties. Interestingly, the conversion of polycyclic aromatic hydrocarbons or their derivatives to fullerenes shares some conceptual similarity to those proposed top-down mechanisms[20].

Here, we show that clusterfullerenes are formed from doped graphite, a universal starting material for carbon nanostructure synthesis, by a laser-based synthesis method that permits in situ formation investigations and uncover unprecedented mechanistic insight into self-assembly of complex carbon compounds. We disclose that the formation of high-symmetry clusterfullerenes occurs through a distinct bottom-up mechanism and that high-yield formation of $Sc_3N@C_{80}$ is achieved by the complete circumvention of Buckminsterfullerene, $C_{60}$[21], in bottom-up reaction paths. To validate our model and probe cage selection and cluster size effects, we directly study the small, non-IPR compound, $Sc_3N@D_3$-$C_{68}$, and larger, high-symmetry species, including $M_3N@I_h$-$C_{80}$ (M = group 3 metal), under precise synthetic conditions.

## Results

**$M_3N$-encapsulated (M = group 3) cages from doped graphite.** To devise a strategy to study in situ self-assembly processes for these nanomaterials, we performed extensive analyses by laser vaporization of graphite-based starting materials doped with group 3 metal oxides and numerous sources of heteroatoms, such as gaseous (e.g., ammonia and $N_2$) and molecular nitrogen sources. In fact, these heteroatom sources are used in arc discharge synthesis to produce macroscopic quantities of the compounds[22]. In our approach, online chemical sampling is carried out by use of a pulsed laser vaporization cluster source, analyzed by state-of-the-art Fourier transform ion cyclotron resonance (FT-ICR) mass spectrometry, which was previously limited to

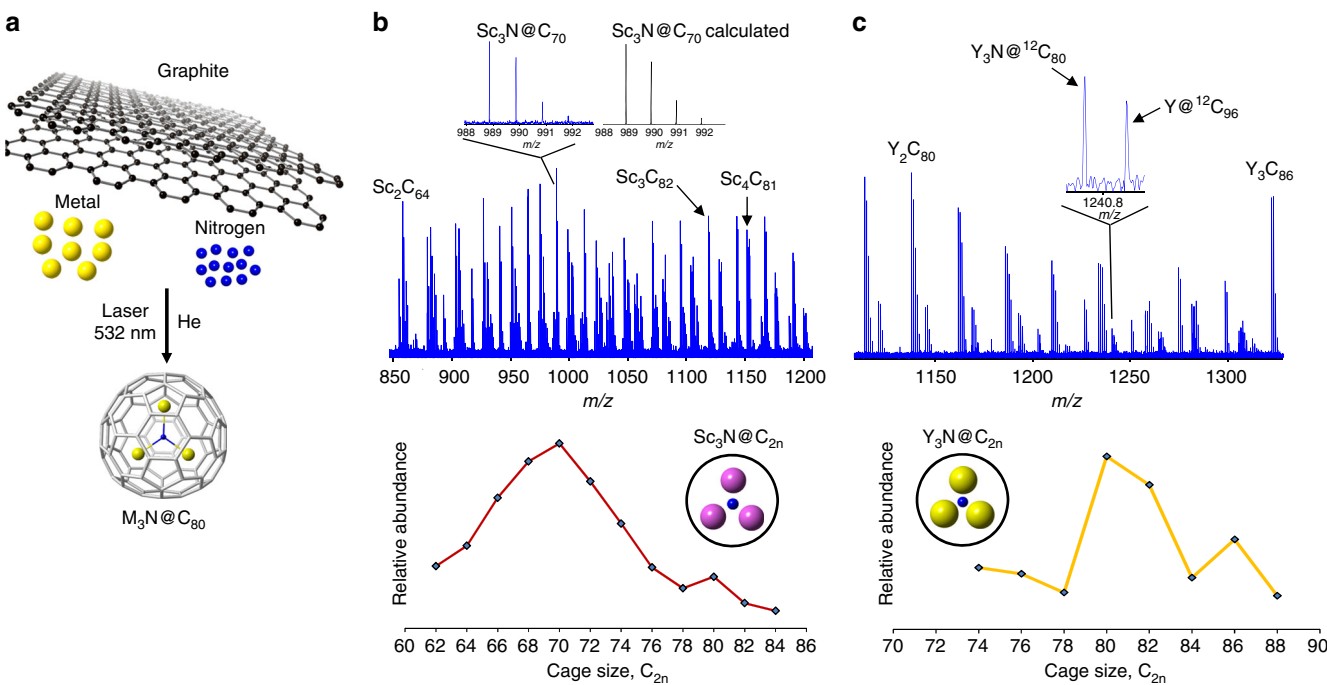

**Fig. 1** Clusterfullerenes formed by laser vaporization of group 3 metal-doped and nitrogen-doped graphite. **a** Synthesis schematic for clusterfullerenes formed from a mixture of graphite, metal oxide, and melamine (nitrogen source) in this work. FT-ICR mass spectra of cluster cations generated by laser vaporization of **b** Sc-doped and N-doped graphite and **c** Y-doped and N-doped graphite. $M_3N@C_{2n}$ formation distributions are graphically shown below each spectrum

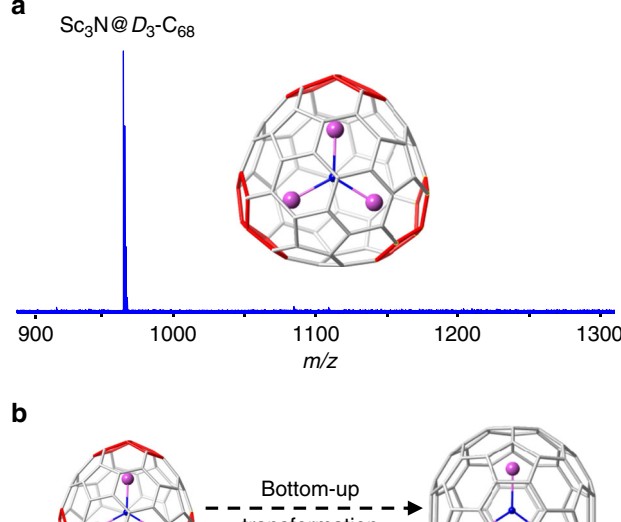

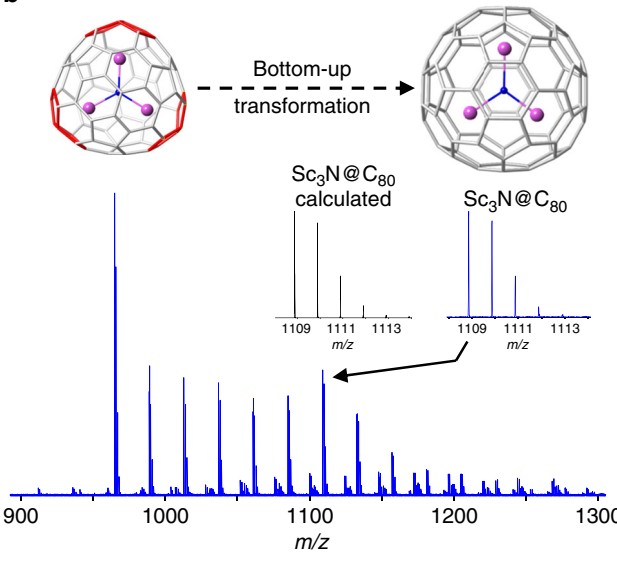

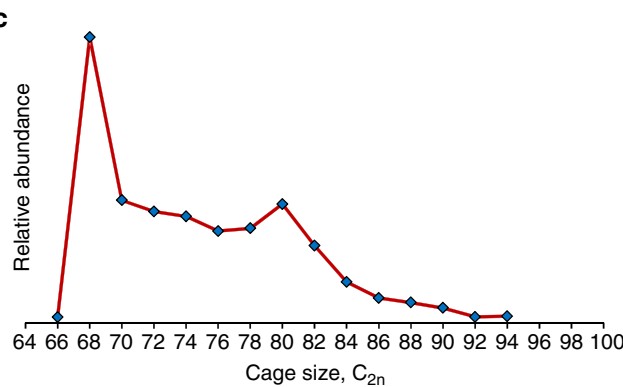

**Fig. 2** Bottom-up growth of a small, fused pentagon-containing clusterfullerene. **a** Low energy (~2 mJ) laser desorption spectrum (positive ions) of isomerically pure, $Sc_3N@D_3-C_{68}$, without exposure to carbon vapor from graphite. **b** Molecular reactivity and behavior of $Sc_3N@C_{68}$ in carbon vapor from graphite in a He atmosphere (~10 mJ per pulse). **c** Growth distribution for $Sc_3N@C_{68} + nC_2$

empty cages and conventional metallofullerenes[23–25]. We find that the solid organic nitrogen sources[26] are an excellent choice for the formation of trimetallic NCFs under the present laser-based conditions and, in particular, melamine yields highly reproducible formation products[27, 28]. In these strongly ionizing environments, positive ions are expected to be representative of the neutral NCF distributions, similar to empty cages and

mono-metallofullerenes[23]. Figure 1 shows molecular nanocarbon products formed by laser vaporization (532 nm, 10 mJ per pulse) of a stationary target rod comprised of graphite, scandium oxide, and melamine (10% atom Sc, 1:2 ratio for $Sc:C_3H_6N_6$) in a He atmosphere. Surprisingly, the small compounds, such as $Sc_3N@C_{68}$, $Sc_3N@C_{70}$, $Sc_3N@C_{72}$, and $Sc_3N@C_{74}$, exhibit higher relative abundance than $Sc_3N@C_{80}$ and similar sized cages. That $Sc_3N@C_{80}$ nanocluster, however, displays an enhanced abundance compared to other medium-sized cages and is a "magic numbered" species. Under the present clusterfullerene-generating conditions, empty cage fullerenes are suppressed by laser synthesis, similar to observations for arc discharge synthesis from doped graphite containing these particular starting materials. Notably, all $Sc_3N@C_{2n}$ ($C_{2n} = 68, 70, 78, 80, 82$) synthesized and isolated by means of the arc discharge methods correspond to the observed cage sizes[2, 29].

Although the smallest species, $Sc_3N@C_{62}-C_{66}$, are formed in abundance under the present conditions, they may "react away" in the solid state or upon exposure to air or solvent. Therefore, some of these pristine isomers may not be readily detectable in arc discharge extracts or soot, as for other smaller non-IPR fullerenes[30]. Another possibility is that more carbon vapor may be available for insertion reactions in arc discharge than for the present laser vaporization conditions. However, it is clear that the $Sc_3N@C_{80}$ clusterfullerene is an abundantly produced medium-sized $Sc_3N@C_{2n}$ formed by laser vaporization of doped graphite. At lower laser fluence (~5 mJ), we find that the smallest clusterfullerene cages, $Sc_3N@C_{2n}$ ($C_{2n} = C_{62}$ to ~$C_{74}$), are formed but medium-sized species, such as $Sc_3N@C_{80}$, are not observed. These results show that the smallest $Sc_3N$-based clusterfullerenes appear to form before the larger compounds from doped graphite under our conditions.

Structural analysis of $Sc_3N@C_{2n}$ produced from the bulk graphite-based starting material is obtained by collision-induced dissociation (CID) experiments, performed by means of sustained off-resonance irradiation (SORI)[31]. Supplementary Fig. 1 identifies the fragmentation pattern of gas-phase isolated $Sc_3N@C_{70}$ formed by laser vaporization of the graphite/$Sc_2O_3$/melamine mixture. The singular dissociation pathway observed for $Sc_3N@C_{70}$ in an ultrahigh vacuum is a $C_2$-elimination event. The internally bound cluster, $Sc_3N$, remains trapped within the nanoscale void of the carbon cage when highly thermally excited by collisions, whereas any exohedrally bound metals or heteroatoms would readily dissociate. Therefore, dissociation investigations provide compelling evidence that $Sc_3N@C_{70}$ exhibits a NCF structure. Larger $Sc_3N@C_{2n}$ formation products display that same characteristic clusterfullerene dissociation pattern.

To discern the influence of the $M_3N$ cluster size effect with respect to the step of initial cage nucleation by means of bottom-up formation from starting material plasma, Y-doped melamine-containing graphite is examined under identical conditions. The larger ionic radius of Y (0.90 Å), and thus larger $M_3N$ cluster, compared to Sc (0.75 Å), provides a mechanistic avenue to experimentally probe that process. A striking change in formation distribution is observed for $Y_3N@C_{2n}$ compared to $Sc_3N@C_{2n}$ (Fig. 1). The small $M_3N$-based clusterfullerenes observed for $Sc_3N@C_{2n}$ are entirely absent, and, instead, nanocarbons are shifted to cage sizes of $Y_3N@C_{74}$ to $Y_3N@C_{88}$[32]. However, the $Y_3N@C_{80}$ molecular ion, like $Sc_3N@C_{80}$, exhibits higher relative abundance indicating formation of a stable $C_{80}$ isomer. The $Y_3N@C_{2n}$ compounds, for example, $Y_3N@C_{80}$, are confirmed to be endohedral NCFs by SORI-CID investigations, as expected. Other families of metallofullerenes are detected from both Sc-doped and Y-doped carbon plasma systems and correspond to $M_2C_{2n}$, $M_3C_{2n}$, $M_4C_{2n}$, as well as odd carbon-numbered clusters

such as $M_3C_n$ that must contain at least one C within a cage[33]. Thus, in addition to the successful formation of NCFs, numerous forms of clusterfullerenes are also produced from doped graphite by laser vaporization. Importantly, the observed cage size shift for $Sc_3N$ compared to $Y_3N$ clarifies that the $M_3N$ cluster nucleates formation of the smallest cages in the first bottom-up step.

$Sc_3N@C_{2n}$ are formed in higher relative abundance than $Y_3N@C_{2n}$ generated by laser vaporization of the starting materials, and therefore $Sc_3N$ is more efficiently entrapped within cages under our conditions. Thus, Sc appears to possess a special ability to bond with C and/or N in the initial nucleation step. Moreover, molecular ions that contain up to four Sc atoms are readily observed from Sc/N/C condensing plasma, whereas at most three Y atoms can be encapsulated from the Y/N/C plasma under the same vaporization parameters. These results provide mechanistic insight into how Sc may "pull in" other elements into cages through initial formation of the smallest NCFs[34], which we propose grow into larger species through bottom-up reactions as we clearly demonstrate below.

**Transformation of $Sc_3N@D_3$-$C_{68}$ into $Sc_3N@C_{80}$.** Chemical processes involved in the growth of the initially formed, small $Sc_3N$-based clusterfullerenes from bulk starting materials are elucidated by analysis of isomerically pure NCF cages in carbon vapor generated from graphite, conducted under the same high energy formation conditions (10 mJ per pulse). These studies also provide insight into structural effects with respect to nanocarbon reactions that operate during self-assembly. Figure 2 shows that $Sc_3N@D_3$-$C_{68}$ unambiguously exhibits bottom-up growth by carbon insertion reactions that results in formation of larger $Sc_3N@C_{2n}$ ($C_{2n} = 70$–94) compounds. Under the present conditions, the most abundant growth product is $Sc_3N@C_{80}$, which exhibits an enhanced abundance compared to other medium-sized cages, similar to $Sc_3N@C_{80}$ generated by laser vaporization of bulk doped graphite (Fig. 1). Thus, in both cases, the results indicate that a stable, high-symmetry $C_{80}$ cage isomer is formed. Dissociation investigations confirm that $Sc_3N$ is encapsulated in clusterfullerene growth product cages (Supplementary Figs. 2, 3).

To distinguish formation mechanisms for the small $Sc_3N@C_{2n}$ ($C_{2n} = 62$–66) products synthesized by laser vaporization of Sc-doped and N-doped graphite, which have not been detected in extracts or soot from arc discharge, $Sc_3N@D_3$-$C_{68}$ is studied in the absence of graphite vapor under the high energy conditions of synthesis. Supplementary Fig. 4 shows products generated from $Sc_3N@D_3$-$C_{68}$ by direct laser ablation (10 mJ per pulse) without exposure to carbon vapor. Surprisingly, $Sc_3N@C_{70}$, a bottom-up growth structure, is the most abundant molecular reaction product, suggesting that carbon insertion reactions can be favored in low carbon density, high-energy conditions. $Sc_2N@C_{68}$, a Sc-loss product, and $Sc_3N@C_{66}$, formed by $C_2$-loss, are observed only in very low abundance. Consequently, these results provide additional evidence that the smallest $Sc_3N@C_{2n}$ ($C_{2n} = 62$–66) are not formed from larger $Sc_3N@C_{2n}$ by top-down processes during self-assembly from graphite starting material and are consistent with a bottom-up formation mechanism.

Reaction products that correspond to non-NCF compounds, namely, $Sc_3NC_n$, $Sc_2NC_n$, and $ScNC_n$ ($C_n$ = odd number of C atoms) also result after growth of $Sc_3N@D_3$-$C_{68}$ in carbon vapor (Supplementary Fig. 5). We find that all odd-numbered carbon chemical compositions dissociate by $C_2$-elimination with retention of an odd number of carbon and all Sc and N atoms. Consequently, the presence of a carbon adatom is excluded because it readily dissociates at low thermal energy. Therefore, all of these molecular products contain clusters of Sc, N, and C

within cages. As a result of the present high energy formation conditions that render compounds with no C adatoms, the $Sc_3NC_n$ species plausibly exhibit structures, $Sc_3NC@C_{2n}$, in which the five-atom cluster, $Sc_3NC$, is entrapped within even-numbered carbon cages. These species result from intramolecular reactions that take place during the growth process. Notably, the smallest member of $Sc_3NC_n$ observed from $Sc_3N@D_3$-$C_{68}$ growth is $Sc_3NC_{81}$ (Supplementary Fig. 5), which may be a contributing formation route to $Sc_3NC_{81}$ that has been isolated by means of arc discharge and exhibits a carbonitride clusterfullerene structure, $Sc_3NC@C_{80}$[35].

**Computational analysis for reaction paths from $C_{68}$ to $C_{80}$.** $Sc_3N@D_3$-$C_{68}$ is clearly demonstrated to transform into $Sc_3N@C_{70}$ after exposure to carbon vapor generated from graphite, which involves the overall incorporation of $C_2$ into the caged network. That nanocluster is also among the most abundant NCFs formed by laser vaporization of Sc-doped and N-doped graphite in this work. Interestingly, the only $Sc_3N@C_{70}$ species that has been isolated is $C_{70}(7854)$[36]. Note that we will now use the spiral algorithm numerical identifier in parenthesis to distinguish isomers for a particular cage size[37]. To discern a reaction path for the $Sc_3N@D_3$-$C_{68}$ to $Sc_3N@C_{70}$ transformation, all possible topological $C_2$ insertions were analyzed for the $D_3$-$C_{68}(6140)$ cage in the hexaanionic form, $C_{2n}^{6-}$, to account for charge transfer from the encapsulated $Sc_3N$ cluster to the cage. Six isomers of $C_{70}$ can be generated by direct $C_2$ insertion (Supplementary Fig. 6) without the involvement of Stone-Wales (SW) bond rearrangements. The two lowest energy product structures, shown in Fig. 3a, b, are found to be associated with very exothermic energies (Supplementary Table 1). $C_{70}(7886)$ exhibits a classical structure comprised of pentagons and hexagons, whereas $C_{70}(hept)$ possesses a non-classical structure that contains a heptagon motif.

Analysis of all possible $C_2$ insertions into these predicted $C_{70}$ isomers (Supplementary Fig. 7) is performed for the next bottom-up cage transformation and plausible $C_{72}$ intermediates are found to be $C_{72}(10611)$, $C_{72}(10610)$, and $C_{72}(hept1)$, as shown in Fig. 3c. Notably, $C_{72}(10611)$ acts as a "gateway" cage and is intimately related by a single SW rearrangement (Supplementary Fig. 8) to $C_{72}(10610)$ with a barrier (~150 kcal mol$^{-1}$) that can be surpassed at the temperature of fullerene formation (>1000 K). Although the heptagon structure is 24.2 kcal mol$^{-1}$ higher in energy than $C_{72}(106011)$, we find that the energy profiles for these cage growth transformations are similar (Supplementary Fig. 9). Therefore, formation of classical and heptagon-containing cages should take place during bottom-up clusterfullerene growth, which is in agreement with the recent characterization of trimetallic nitride, $LaSc_2N@C_{80}(hept)$, as well as for larger heptagon-containing cage sizes[15, 38, 39]. Structures that possess two heptagons are not considered because such cages are very strained and are at least 54 kcal mol$^{-1}$ higher in energy than $C_{72}(10611)$. Reaction energies for all of the lowest energy isomers were computed in the clusterfullerene form for comparison to the hexaanionic investigations. Supplementary Table 2 shows that reaction energies are somewhat lower, but remain very exothermic. Thus, formation of the proposed $C_{70}$ and $C_{72}$ growth structures is favorable based on thermodynamic considerations. A detailed description of the growth mechanism is given in Supplementary Fig. 9.

We have investigated global pathways to high-symmetry $C_{80}$ clusterfullerenes from $Sc_3N@D_3$-$C_{68}$ based on this strategy for step-by-step cage formations of low energy isomers through $C_2$ insertions and bond rearrangements. Despite the complexity of the processes involved (Supplementary Fig. 10), it is

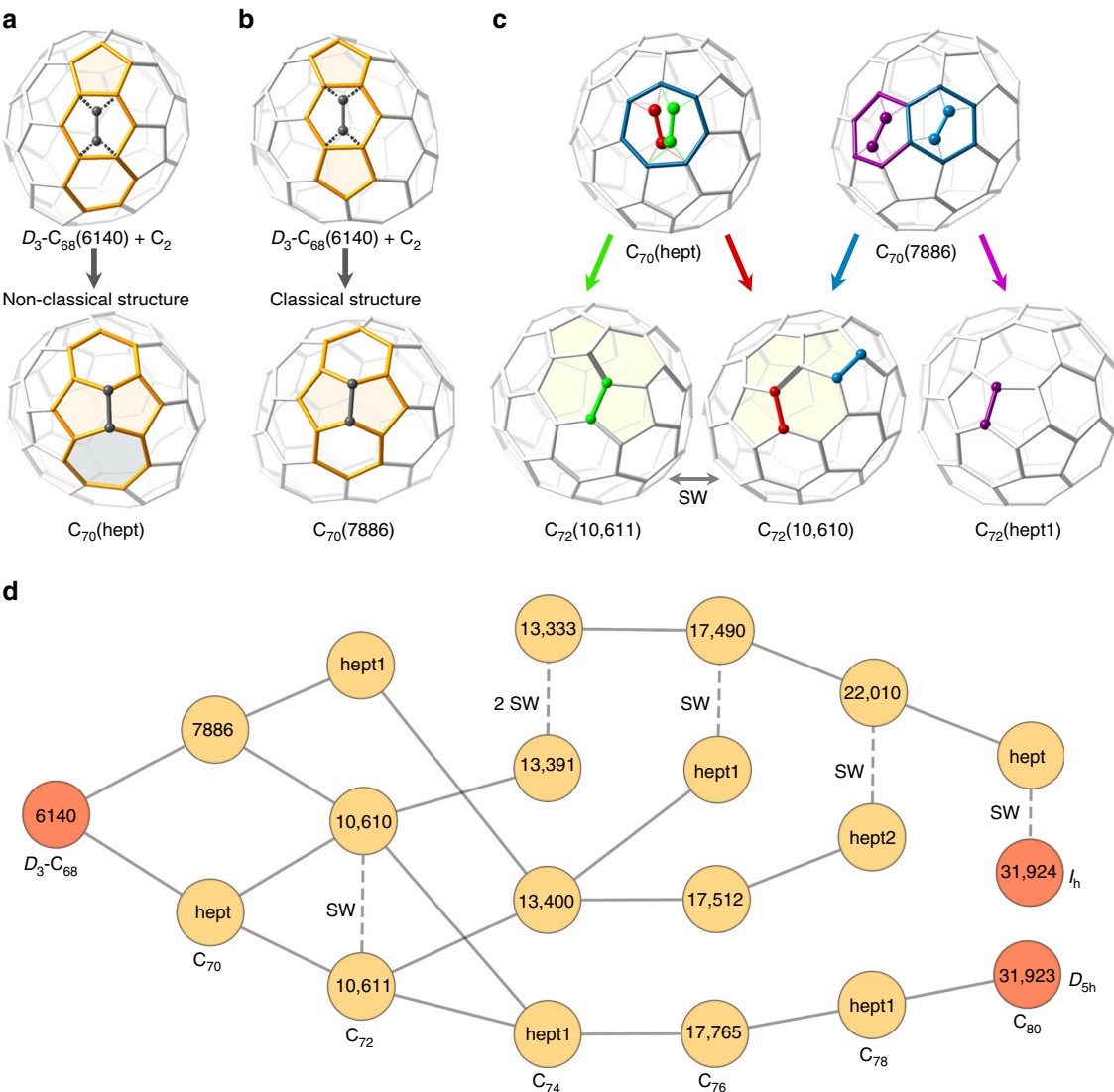

**Fig. 3** Reaction paths to high-symmetry $C_{80}$ isomers ($I_h$, $D_{5h}$) from $D_3$-$C_{68}$ through non-classical and classical cages. $D_3$-$C_{68}$ can grow by $C_2$ insertion into the **a** non-classical structure that contains a heptagon motif, $C_{70}$(hept), or **b** a classical structure comprised of only pentagons and hexagons, $C_{70}$(7886). **c** For the next bottom-up cage transformation, $C_{70}$ to $C_{72}$, the most plausible isomers are found to be the classical cages, $C_{72}$(10611), $C_{72}$(10610), and heptagon-containing $C_{72}$(hept1). **d** From those $C_{72}$ isomers, reaction paths to high-symmetry $C_{80}$ cages, $I_h$-$C_{80}$ and $D_{5h}$-$C_{80}$, involve $C_2$ insertion reactions and two to three SW rearrangements through classical and non-classical cage intermediates

extraordinary that several relatively simple pathways exist and are shown in Fig. 3d. Initiating from the proposed $C_{68}$ to $C_{72}$ structure progression, we find reaction paths from $D_3$-$C_{68}$ to icosahedral $C_{80}$ that involve the known NCF cages, $C_{78}$(22010), $C_{80}$(hept), and $C_{80}$($I_h$) in final reaction sequence, where $C_{80}$(hept) is the structure of recently isolated NCF, LaSc$_2$@$C_{80}$(hept)[15, 40–42]. Strikingly, Fig. 3d also shows that a bottom-up sequence of six direct $C_2$ insertions without any SW rearrangement results in the known high-symmetry $D_{5h}$-$C_{80}$ cage, M$_3$N@$D_{5h}$-$C_{80}$(31923). Thus, we find that all plausible bottom-up growth routes to high-symmetry $C_{80}$ involve a total of six $C_2$ insertion reactions and two to three $C_2$ rearrangements. Importantly, reaction energies, and free energies computed at the high temperature of synthesis (2000 K), for all routes are confirmed to be very exothermic (Supplementary Tables 2–4). It is possible that some of these predicted non-classical and classical intermediates may be isolated and characterized in the near future (Fig. 3, Supplementary Fig. 10, and Supplementary Table 5).

**Growth of M$_3$N@$C_{80}$ into larger, lower symmetry cages**. As a crucial test for our proposed bottom-up formation mechanism from graphite, the M$_3$N@$I_h$-$C_{80}$ compound is specifically investigated under the harsh conditions typical of plasma synthesis, thereby providing important nanocarbon mechanistic information because of its icosahedral symmetry and medium cage size. Figure 4a–c shows the products formed after exposure of isomerically pure Sc$_3$N@$I_h$-$C_{80}$ to carbon plasma in He. An identical laser fluence (10 mJ per pulse) is used for all molecular reactivity studies in this work to facilitate comparison to gas-phase synthesis of Sc$_3$N@$C_{2n}$ from raw starting materials without pre-existing NCFs. Numerous larger metallic nitride species are formed by repeated carbon insertion reactions into the Sc$_3$N@$I_h$-$C_{80}$ precursor. The dominant nanocarbon reaction after exposure of Sc$_3$N@$I_h$-$C_{80}$ to graphite vapor is a single $C_2$ insertion to yield Sc$_3$N@$C_{82}$. By contrast, species smaller than Sc$_3$N@$C_{80}$ are absent, including Sc$_3$N@$C_{68}$, revealing that carbon loss or top-down formation to smaller cages is not a significant process under the high energy, high carbon density conditions of

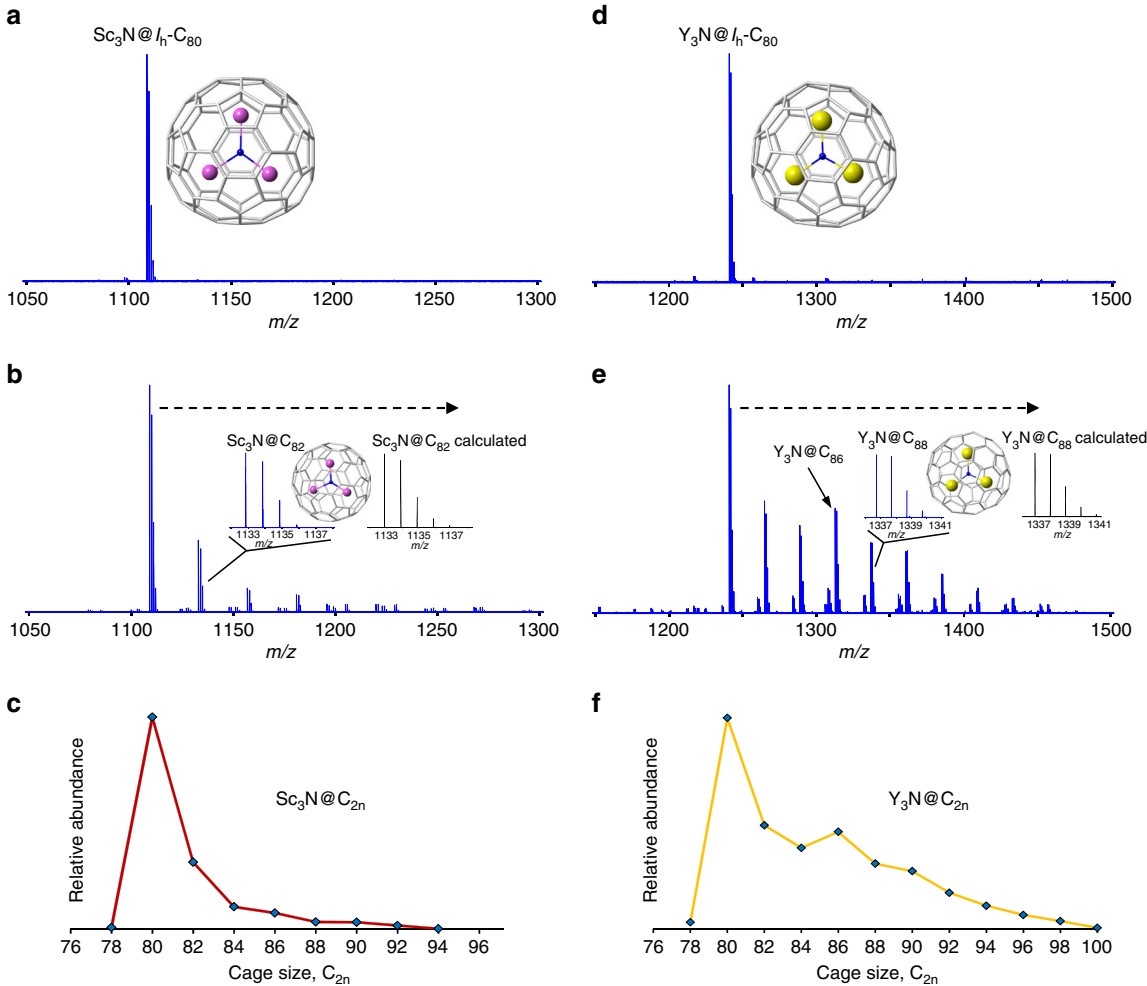

**Fig. 4** Influence of the encapsulated cluster on growth of icosahedral $C_{80}$. Isomerically pure **a** $Sc_3N@I_h$-$C_{80}$ after laser desorption (~2 mJ) without carbon vapor and **b** after reaction with graphite vapor in He (10 mJ) and **c** $Sc_3N@C_{2n}$ + $nC_2$ formation distribution. Comparison of isomerically pure **d** $Y_3N@I_h$-$C_{80}$ after laser desorption (~2 mJ) and **e** reaction with graphite vapor under identical conditions (10 mJ) and **f** $Y_3N@C_{2n}$ + $nC_2$ formation distribution. $C_2$-elimination events are not readily observed for either $M_3N@I_h$-$C_{80}$

synthesis. The structures of the molecular product ions $Sc_3N@C_{2n}$ ($C_{2n} \geq 82$) formed through bottom-up growth of $Sc_3N@I_h$-$C_{80}$ are confirmed (Supplementary Fig. 11) to be trimetallic NCFs. In addition, we find that carbon loss reactions do not readily take place when $Sc_3N@I_h$-$C_{80}$ is subjected to high energy laser ablation (10 mJ per pulse) without carbon vapor, i.e., high energy, low carbon density conditions (Supplementary Fig. 12), a molecular behavior consistent with the much smaller $Sc_3N@D_3$-$C_{68}$. The $D_{5h}$-$C_{80}$ isomer encapsulated by $Sc_3N$ was also probed in graphite vapor and exhibits bottom-up growth behavior, and it appears to be slightly more reactive than $I_h$-$C_{80}$ (Supplementary Fig. 13).

To probe cluster-cage size effects with respect to carbon insertion reactions for the $I_h$-$C_{80}$ cage, molecular reactivity studies are performed whereby $Y_3N$ is substituted for $Sc_3N$. Figure 4d–f shows products that result after exposure of isomerically pure $Y_3N@I_h$-$C_{80}$ to graphite vapor, conducted under identical conditions to those for the $Sc_3N@I_h$-$C_{80}$ (and $Sc_3N@D_3$-$C_{68}$) growth experiments. $Y_3N@C_{82}$, formed by $C_2$ incorporation, is formed in high relative abundance; however, $Y_3N@C_{84}$ and $Y_3N@C_{2n}$ compounds as large as $Y_3N@C_{100}$ are present in substantial abundance. In addition, $Y_3N@I_h$-$C_{80}$, like $Sc_3N@I_h$-$C_{80}$, does not readily exhibit carbon loss events or top-down behavior (Supplementary Fig. 14) when subjected to high energy laser ablation in absence of carbon vapor (i.e., low carbon

density, high energy conditions). Extensive SORI-CID experiments strongly support that the major growth products are homogeneous Y-based NCFs (Supplementary Fig. 15). The more extensive bottom-up growth of $Y_3N@I_h$-$C_{80}$ is in agreement with the formation distribution of $Y_3N@C_{2n}$ generated from Y-doped and N-doped graphite (Fig. 1). Thus, the bottom-up mechanism is attributed to also operate in the Y-containing graphite starting material plasma.

Further evidence to corroborate metallic NCF structures for $Y_3N@C_{2n}$ growth products, as well as $Sc_3N@C_{2n}$, synthesized in this work is obtained by dissociation analysis of the endohedral heterofullerene, $Y_2@C_{79}N$, which exhibits an 80 atom cage that substitutes N for a single C atom in the caged network[43]. We find that the dissociation route for $Y_2@C_{79}N$ is CN-elimination (Supplementary Fig. 16), in contrast to $C_2$-loss with retention of N for the $Y_3N@C_{80}$ endohedral nanocluster. These results are consistent with empty cage N-containing heterofullerenes[44]. Thus, the present gas-phase dissociation studies are shown to be a structural diagnostic for identification of clusterfullerenes and endohedral heterofullerenes.

The mechanistic uniqueness of $Sc_3N@I_h$-$C_{80}$ is clearly established in this work by its lack of growth into larger species, whereas $Y_3N@I_h$-$C_{80}$ clearly diverges from that trend and can grow into cages that exceed 100 carbon atoms in size. These

results also support the assignment of a high-symmetry cage, such as $I_h$-$C_{80}$, as a major contributing isomer for the magic numbered $Sc_3N@C_{80}$ nanocluster formed from bulk starting materials by laser synthesis, although other isomers should be produced to a lesser extent. Pyramidalization of carbon atoms takes place when a metal is coordinated to the [5,6] carbon atoms for $Sc_3N@C_{80}$ compared to the center of a hexagon for $Y_3N@C_{80}$[45]. Consequently, the pyrene-type carbon atoms in $Y_3N@C_{80}$ are more strained and may be more reactive to carbon insertion reactions. It is noteworthy that the $M_3N$-encapsulated product cages shown in Fig. 4 must be lower symmetry than the $I_h$-$C_{80}$ precursor. Interestingly, the small non-IPR $Sc_3N@D_3$-$C_{68}$ compound that contains three sets of fused pentagons appears to be somewhat less reactive in the bottom-up growth scheme than $Y_3N@I_h$-$C_{80}$. That result distinguishes how the transfer of six electrons and an optimally sized $M_3N$ cluster significantly renders non-IPR cages less reactive through bottom-up mechanisms. Those observations further account for the different formation trends for $Sc_3N@C_{2n}$ and $Y_3N@C_{2n}$ from the original doped graphite.

## Discussion

We propose that high-symmetry clusterfullerenes primarily self-assemble through a bottom-up mechanism by simple vaporization of graphite. Online chemical sampling of laser vaporized doped graphite reveals an initial nucleation step, whereby $M_3N$ nucleates formation of the smallest possible cage(s). For example, $Sc_3N$ bypasses cages smaller than $C_{62}$, whereas the larger $Y_3N$ cluster must initially nucleate cages of $C_{74}$ or larger under our conditions. Those results clarify that formation initiates through bottom-up processes that exhibit a cluster size effect on cage selection. Avoidance of $I_h$-$C_{60}$ in the bottom-up mechanism and thus the many other small "bottleneck" cages, $C_{28}$, $C_{36}$, $C_{44}$, $C_{50}$, in reaction paths that severely limit growth to medium-sized conventional metallofullerenes, $M@C_{2n}$[23, 46], enable clusterfullerene cages in the solvent-extractable ~$C_{70}$–$C_{100}$ region to form in inherently higher abundance. Further, that mechanistic property allows breakthrough access to the next possible icosahedral cage, $I_h$-$C_{80}$, which may then act as a mechanistic bottleneck in formation for group 3 NCFs, and, in particular, should permit $Sc_3N@I_h$-$C_{80}$ to accumulate in bottom-up reaction paths and thus explains its high-yield formation.

Our bottom-up model for NCF formation is further experimentally tested by extensive investigations on specified isomers of the group 3-based clusterfullerenes, $Sc_3N@D_3$-$C_{68}$, $Sc_3N@I_h$-$C_{80}$, $Sc_3N@D_{5h}$-$C_{80}$, and $Y_3N@I_h$-$C_{80}$ by exposure to graphite vapor under characteristic physicochemical synthetic conditions. Fused pentagon-containing $Sc_3N@C_{68}$ unambiguously grows into $Sc_3N@C_{80}$, which exhibits an enhanced abundance in agreement with the proposal that medium-sized $Sc_3N@C_{2n}$ from bulk doped graphite primarily form through a bottom-up mechanism. The archetypal clusterfullerene, $Sc_3N@I_h$-$C_{80}$, is observed to be rather inert to further bottom-up growth, which is consistent with the assignment of $I_h$-$C_{80}$ for $Sc_3N@C_{80}$ from doped graphite and from the explicit growth of $Sc_3N@D_3$-$C_{68}$ in this work. Theoretical investigations show that cage transformations from $D_3$-$C_{68}$ cage into $I_h$-$C_{80}$ can occur by a total of six $C_2$ insertions and only two to three SW rearrangements. Heptagon-containing and classical cages are predicted to be involved in these transformations. The $D_{5h}$-$C_{80}$ isomer can self-assemble without any SW rearrangements through six direct $C_2$ insertion events.

Substitution of $Y_3N$ in the $I_h$-$C_{80}$ cage dramatically alters the molecular behavior of $I_h$-$C_{80}$ in graphite vapor and $Y_3N@C_{80}$ readily grows into clusterfullerenes that exhibit 100 carbon atom cages. That cluster-cage size effect further explains the shift to larger cages in formation distributions for $Sc_3N@C_{2n}$ to $Y_3N@C_{2n}$

from doped graphite. Unexpectedly, we find that NCFs can transform into five-atom cluster entrapped cages that presumably result from intramolecular cage reactions that take place during bottom-up growth of $Sc_3N@C_{68}$. For example, endohedral species with a chemical composition of $Sc_3NC_{81}$ are formed that may be the origin of five-atom cluster encapsulated, $Sc_3NC@C_{80}$, and thus offers another mechanistic route to clusterfullerene formation through bottom-up growth paths. We do not observe formation of $M_3NC_{81}$ by growth of the larger $Sc_3N@I_h$-$C_{80}$, $Sc_3N@D_{5h}$-$C_{80}$, or $Y_3N@I_h$-$C_{80}$ clusterfullerenes NCFs under the present conditions, suggesting that cluster interaction with the cage during bottom-up growth could be crucial to its formation.

In conclusion, through in situ investigations of clusterfullerenes synthesized by laser vaporization of doped graphite and study of discrete NCF compounds, combined with extensive theoretical investigations, we disclose that bottom-up self-assembly reactions are responsible for synthesis of high-symmetry carbon cages that encapsulate metallic nitride clusters. We propose that these intrinsic chemical processes are a fundamental property of carbon under harsh conditions typical of synthesis and will help tackle the challenge of carbon nanostructure formation for other hybrid carbon allotropes and should be useful for synthesis of new carbon-based cluster compounds. We note that it is possible that a "local" $C_2$-loss event may occur in a "global" bottom-up path or in the solid state after formation, and therefore should also be considered to comprehensively describe fullerene formation. Future in situ studies of doped graphite, combined with the analysis of distinct clusterfullerenes, are now possible for compounds that encapsulate other endohedral clusters (e.g., metal carbides, sulfides, oxides, etc.)[47–53] and possess various structural motifs, which should facilitate the exploration of new forms of encapsulated nanocarbon materials and their fundamental self-assembly processes.

## Methods

**Clusterfullerenes from doped graphite**. Doped graphite starting materials are produced by physical mixing of graphite (99.9995%, 2–15 μm), scandium oxide or yttrium oxide (99.9%), and melamine (99%)[23]. The doped graphite material is then molded into a 12.7-mm rod by compression of the mixture. Doped graphite rods are comprised of 10 atom % metal, with a 1:2 ratio for metal:$C_3H_6N_6$.

**Reactivity of cluster-encapsulated cages in graphite vapor**. Macroscopic samples of metallic NCFs are synthesized by arc discharge. Isomerically pure samples of $Sc_3N@I_h$-$C_{80}$ and $Sc_3N@D_3$-$C_{68}$ were purified by use of non-chromatographic methods[54]. $Y_3N@I_h$-$C_{80}$ was purified by multi-stage HPLC. Isomerically pure NCF was then uniformly applied to the surface of a pristine 12.7-mm graphite rod (99.9995%, 2–15 μm) for nanocarbon reaction studies by use of a Nd:YAG (532 nm, 10 mJ) pulsed laser cluster source[23, 25]. NCFs were individually applied to a quartz rod for high energy, direct laser ablation without exposure to carbon plasma.

**Cluster source and 9.4 T FT-ICR mass spectrometry**. Self-assembly reaction experiments are analyzed with a custom-built FT-ICR mass spectrometer based on a 9.4 T superconducting magnet and performed with positive ions produced by a pulsed laser cluster source[23, 24]. Evaporation of a doped graphite stationary rod (12.7-mm diameter) is achieved by a single laser shot fired from a Nd:YAG (532 nm, 3–5 ns pulse width, 1.5-mm beam diameter) in conjunction with the opening of a pulsed valve (800 ms duration) to admit He flow over the sample. Carbon vapor produced then enters a channel 4 mm in diameter and ~8.5 mm in length. The laser is fired ~2 ms after opening of the pulsed valve for evaporation of doped graphite samples and ~4 ms for $Sc_3N@C_{68}$, $Sc_3N@I_h$-$C_{80}$, $Sc_3N@D_{5h}$-$C_{80}$, and $Y_3N@I_h$-$C_{80}$-coated graphite samples. Ions accumulated by ten individual laser and helium pulse events are transported to an open cylindrical ion trap (70-mm diameter, 212-mm long, aspect ratio ~2). The ions are accelerated to a detectable cyclotron radius by a broadband frequency sweep excitation (260 Vp-p, 150 Hz μs$^{-1}$, 3.6 down to 0.071 MHz) and subsequently detected as the differential current induced between two opposed electrodes of the ICR cell. Each of the acquisitions is Hanning-apodized and zero-filled once before fast Fourier transform and magnitude calculation[23]. Up to ten time-domain acquisitions are averaged. The experimental event sequence is controlled by a modular ICR data acquisition system. Ions are further probed by CID.

**Computational details**. Amsterdam Density Functional code (ADF2012) was used for the electronic structure calculations[55]. The electronic density was provided by the local density approximation by use of Becke's gradient corrected exchange functional, and Vosko, Wilk, Nusair parametrization for correlation, corrected with Perdew's functional (BP86). Electrons for all the atoms were described with Slater-type basis functions of triple-$\zeta$ + polarization quality. We have included scalar relativistic corrections by means of the zeroth-order regular approximation formalism. All $Sc_3N@C_{2n}$ calculations have also been performed including dispersion corrections. We used the CaGe program to generate fullerenes and Schlegel diagrams.

**Data availability**. Data that support the findings of this study are available within the paper and its supplementary information files, and available from the corresponding author upon request. A data set collection of computational results is available from the online ioChem-BD repository and can be accessed via https://doi.org/10.19061/iochem-bd-2-16, where all intermediates described in Fig. 3 and Supplementary Fig. 10 can be found.

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

## Acknowledgements

Work performed at the National High Magnetic Field Laboratory is supported by the NSF Cooperative Agreement through DMR-11-57490 and the State of Florida. P.W.D. thanks the FSU Research Foundation for support. This work was also supported by the Spanish Ministerio de Ciencia e Innovación (CTQ2014-52774-P) and the Generalitat de Catalunya (2014SGR-199 and XRQTC). L.A. thanks the GC for a predoctoral fellowship (FI-DGR 2014). L.E. thanks the NSF for generous support under PREM program (DMR 1205302) and CHE-140885, and the Robert A. Welch Foundation (Grant AJ-0033). We thank Prof. Gunnar Brinkmann for helpful assistance in the use of CaGe software. We thank John Quinn and Greg Blakney for instrument assistance, and Chad Weisbrod for discussion. Dedicated to the memory of Harry Kroto.

## Author contributions

P.W.D., A.R.-F., J.M.P., and L.E. designed the research and wrote the paper; M.M.-G. performed the experiments under the supervision of P.W.D.; L.A., A.R.-F., and J.M.P. designed and performed the theoretical calculations; M.R.C. and E.C. synthesized and purified samples used in the experiments; M.M.-G., L.A., A.G.M., A.R.-F., L.E., J.M.P., and P.W.D. analyzed data. All authors discussed the results and commented on the manuscript.

## Additional information

**Competing interests:** The authors declare no competing financial interests.

