## [Peer Review File · Nature Communications]

Reviewers' comments:

Reviewer #1 (Remarks to the Author):

The author disclosed chemical processes that result in formation of the endohedral clusterfullerenes such as Sc₃N@C₈₀ etc., establishing the molecular origin and mechanism that underlie formation of unique carbon cage materials. This work combines experimental results and theoretical simulations, and the conclusions are convincing. After several points being clarified, this manuscript is suitable to be published in Nat. Commun.

1. Only using the mass spectroscopy to illustrate the relative abundance of clusterfullerenes is convenient but not accurate, as shown in Figure 1b, the Sc₃N@C₇₀ has the highest yield, that is not consistent with the well-known high yield of Sc₃N@C₈₀ in arc-discharge technique. I wonder whether the trend of clusterfullerenes' abundance would change if the laser energy was gradually increased.
2. While melamine being used as the N-source, I guess the H-element in melamine would also influence the formation of fullerene or metallofullerene.
3. To verify the universality of this technique for clusterfullerene producing, the authors should check up Sc@C_{2n} and Sc₄C₂@C_{2n} family and cite relevant literatures.

Reviewer #2 (Remarks to the Author):

In this contribution, the authors investigated the in-suit growth of metal nitride clusterfullerene under the carbon clustering conditions, which supported the bottom-up formation mechanism. Besides, the authors have studied the influence of encapsulated cluster and cage structure on the reactivity of clusterfullerene in the gas phase, which could be helpful for the understanding the abundance of different clusterfullerene species found in macroscopic synthesis. This work is of fundamental interests in the field of endofullerene chemistry and helpful for understanding the long-sought puzzle about the growth of clusterfullerenes and their growth reactivity in the gas phase. Therefore, I recommend its publication in Nature Communication. Before getting published, however, the following questions should be addressed.

1. The title of this manuscript is unsuitable. This paper focuses on clusterfullerene growth from some specific species such as Sc₃N@c₆₈ and Sc₃N@c₈₀ in the gas phase. The production of endofullerenes from graphite by laser ablation is a quite old topic and has been reported already. It is not the point in the present manuscript. I suggest the author should revise it.
2. The discussion about the cage structure evolution is speculative, as the structural evidence/information on the geometric framework of clusterfullerenes formed in-suit in the gas phase is insufficient. For example, the signal of Sc₃N@c₈₀ found in the gas phase by laser ablation of graphite is hard to be assigned as Sc₃N@Ih-c₈₀ directly. I suggest the authors should tone down to declare the cage structure of formed clusterfullerenes, or additional experiments should be added.
3. In figure 1, the synthetic scheme for clusterfullerene is misleading. The metal and nitrogen source are simply physically mixed with graphite, whereas in this scheme it hits the intercalated graphite complex. It should be revised.

Reviewer #3 (Remarks to the Author):

The authors are submitting findings that address a very serious question in the field of endohedral metallofullerenes. For several decades there has been an ongoing debate regarding either a "bottom-up" or a "top-down" mechanism for their synthesis. This submission presents its findings

to be consistent with a "bottom up" approach. Historically, metallofullerenes have been mostly created by either the (1) laser method [this paper] or by using the (2) electric-arc approach. The experimental and "bottom up" findings for this work are obtained by using the laser approach, a method which may or may not relate to what happens during the arc-plasma synthesis of the electric-arc approach. That is a key question. Would these two methods produce the same bottom-up findings? To what extent are these two methods relatable with their mechanisms of formation? On page 2, top paragraph,*empty cage fullerenes are suppressed, and therefore, we find an inverse formation relationship between empty fullerenes and NCFs by laser synthesis. Notably, all Sc₃N@C_{2n} (C_{2n} = 68, 70, 78, 80, 82) synthesized and isolated by means of the arc discharge methods correspond to the observed cages sizes*.....In response to this paragraph, it may or may not be relevant that an electric-arc paper (Figure 3, JACS, 129, 16257-16259, 2017) also found an inverse formation of C₆₀ and the NCF Sc₃N@C₈₀. In terms of other empty-cage fullerenes, an inverse formation of Sc₃N@C₈₀ NCF and non-C₆₀ fullerenes was observed only at very high additive levels (80%). The only reason for mentioning this reference is to convey that the laser approach and arc reactor are inherently different, but they could still be similar with regard to the bottom-up mechanism and the laser findings of this manuscript.

Overall, I find this submission to be highly meritorious in its scientific impact and intellectual contribution. The findings address a haunting question of the mechanism of endohedral metallofullerene formation. The references are well-cited, the science is sound, and the paper is of broad interest.

In summary, I support acceptance of this manuscript. For perspective, this is the most interesting manuscript that I have read in the last 10+ years. These are exciting findings.

Author Responses to Reviewer Comments

Reviewer #1

The author disclosed chemical processes that result in formation of the endohedral clusterfullerenes such as Sc₃N@C₈₀ etc., establishing the molecular origin and mechanism that underlie formation of unique carbon cage materials. This work combines experimental results and theoretical simulations, and the conclusions are convincing. After several points being clarified, this manuscript is suitable to be published in Nat. Commun.

1. Only using the mass spectroscopy to illustrate the relative abundance of clusterfullerenes is convenient but not accurate, as shown in Figure 1b, the Sc₃N@C₇₀ has the highest yield, that is not consistent with the well-known high yield of Sc₃N@C₈₀ in arc-discharge technique. I wonder whether the trend of clusterfullerenes' abundance would change if the laser energy was gradually increased.

Author Response:

On-line chemical sampling of molecular nanocarbons from doped graphite permits analysis of highly strained carbon cages that may otherwise coalesce in the solid state or 'react away' upon exposure to solvent or air; although, bulk product analysis is not applicable as the reviewer notes. The smallest species, Sc₃N@C₆₂-C₆₆, which have not been observed from extracts of soot produced from arc discharge, must contain fused pentagon motifs. One possibility is that these strained species may be produced but are not readily detectable by use of the arc discharge technique. It is noteworthy that the transformation of doped graphite into Sc₃N@C_{2n} by laser vaporization produces the Sc₃N@C₈₀ clusterfullerene species that exhibits an enhanced abundance compared to those of other medium size cages, an observation that is consistent with the high yield formation of Sc₃N@C₈₀ with respect to bulk synthesis techniques. Another possible difference is that more carbon vapor may be available for reaction by arc discharge vaporization, particularly with setups that use core-drilled graphite rods (which possess an outer 'ring' of pure graphite in the vaporization target). In the context of a bottom-up formation mechanism, whereby the smallest cages form first, there must be a point at which no 'new' small species are formed. However, the clusterfullerenes already formed can still undergo carbon insertion reactions (as shown in Fig. 2 and Fig. 4) and grow into larger species. Thus, the presence of more carbon vapor, without formation of 'new' small cages, could lead to a Sc₃N@C_{2n} distribution more similar to that of Fig.2. Further, our molecular behavior analysis of Sc₃N@C₆₈ and Sc₃N@C₈₀ under representative synthetic conditions in the absence of graphite vapor further shows that the smaller species are not generated by 'shrinking' processes.

Indeed, we have studied the formation distributions under different laser fluence. At ~5 mJ, we observe only the smallest cages without medium-sized species, as described below. At ~10 mJ per pulse, the distribution appears whereby Sc₃N@C_{2n} exhibits medium-size cages, with Sc₃N@C₈₀ possessing an enhanced abundance over other medium-size cages (Fig. 1). That formation distribution is maintained up to ~15 mJ per pulse, after which the formation of any form of fullerene cage is significantly reduced.

We attribute that observation to result as a consequence of bulk doped graphite material flaking off of the vaporization target at sufficiently high fluence, rather than becoming vaporized to produce cages. We plan to report the comparison of $M_3N@C_{2n}$ formation for numerous other doped-graphite starting materials in future studies.

The following text was added to the manuscript to further address the reviewer's comment.

“Although the smallest species, $Sc_3N@C_{62-66}$, are formed in abundance under the present conditions, they may ‘react away’ in the solid state or upon exposure to air or solvent. Therefore, some of these pristine isomers may not be readily detectable in arc discharge extracts or soot, as for other smaller non-IPR fullerenes³⁰. Another possibility is that more carbon vapor may be available for insertion reactions in arc discharge than for the present laser vaporization conditions. However, it is clear that the $Sc_3N@C_{80}$ clusterfullerene is an abundantly produced medium-size $Sc_3N@C_{2n}$ formed by laser vaporization of doped graphite. At lower laser fluence (~5 mJ), we find that the smallest clusterfullerene cages, $Sc_3N@C_{2n}$ ($C_{2n} = C_{62}$ to ~ C_{74}), are formed but medium-size species, such as $Sc_3N@C_{80}$, are not observed. These results show that the smallest Sc_3N -based clusterfullerenes appear to form before the larger compounds from doped graphite under our conditions.”

Reviewer #1

2. While melamine being used as the N-source, I guess the H-element in melamine would also influence the formation of fullerene or metallofullerene.

Author Response:

The ratio of hydrogen to carbon is low in the present experiments. We find no clear evidence that the hydrogen produced from melamine vaporization has had an appreciable influence on $M_3N@C_{2n}$ clusterfullerene formation, in line with our previous demonstration that cages can grow in a pure hydrogen gas atmosphere (Ref 25). Consequently, hydrogen would only be expected to significantly disrupt the initial cage nucleation step (i.e., formation of the smallest possible clusterfullerene cage). Future studies of doped graphite with various hydrogen sources and at higher doping ratios should provide additional insight into these systems and into the mechanistic processes, for example, that result in production of $Sc_3CH@C_{80}$ (Ref 51).

Reviewer #1

3. To verify the universality of this technique for clusterfullerene producing, the authors should check up $Sc@C_{2n}$ and $Sc_4C_2@C_{2n}$ family and cite relevant literatures.

Author Response:

We note that we have studied the mono-metallofullerene family, $Sc@C_{2n}$, formation previously (Ref 23). The investigation of other forms of clusterfullerenes and cluster-encapsulated molecular nanocarbon are now open for study and are expected to yield further mechanistic information, as the reviewer points out.

The following sentence was added to the manuscript (with representative Sc-containing, C_{80} -based clusterfullerenes cited) to address this point.

“Future *in situ* studies of doped graphite, combined with the analysis of distinct clusterfullerenes, are now possible for compounds that encapsulate other endohedral clusters (e.g., metal carbides, sulfides, oxides, etc.)⁴⁷⁻⁵³ and possess various structural motifs, which should facilitate the exploration of new forms of encapsulated nanocarbon materials and their fundamental self-assembly processes.”

Reviewer #2

In this contribution, the authors investigated the in-suit growth of metal nitride clusterfullerene under the carbon clustering conditions, which supported the bottom-up formation mechanism. Besides, the authors have studied the influence of encapsulated cluster and cage structure on the reactivity of clusterfullerene in the gas phase, which could be helpful for the understanding the abundance of different clusterfullerene species found in macroscopic synthesis. This work is of fundamental interests in the field of endofullerene chemistry and helpful for understanding the long-sought puzzle about the growth of clusterfullerenes and their growth reactivity in the gas phase. Therefore, I recommend its publication in Nature Communication. Before getting published, however, the following questions should be addressed.

1. The title of this manuscript is unsuitable. This paper focuses on clusterfullerene growth from some specific species such as $\text{Sc}_3\text{N}@c68$ and $\text{Sc}_3\text{N}@c80$ in the gas phase. The production of endofullerenes from graphite by laser ablation is a quite old topic and has been reported already. It is not the point in the present manuscript. I suggest the author should revise it.

Author Response:

We agree that the study of $\text{Sc}_3\text{N}@D_3\text{-C}_{68}$, $\text{Sc}_3\text{N}@I_h\text{-C}_{80}$, $\text{Sc}_3\text{N}@D_{5h}\text{-C}_{80}$, and $\text{Y}_3\text{N}@I_h\text{-C}_{80}$ in this contribution, performed under representative physicochemical synthetic conditions, is a crucial aspect of the work that permits analysis of specific structural isomers. However, we also believe that the transformation of doped graphite into clusterfullerenes by laser synthesis, shown in Fig. 1, is a central finding. Despite the enormous advances in nanoscience over the past two decades since the discovery of $\text{Sc}_3\text{N}@C_{80}$ (Ref 1), an understanding of how planar sheets of carbon mixed with metal and nitrogen spontaneously transform into spherical cages that entrap four-atom clusters by simple vaporization has remained a long-sought goal. We emphasize that this work is the first report of clusterfullerene synthesis from doped graphite by a laser synthesis method. Thus, this work strongly suggests that the fundamental, bottom-up self-assembly processes elucidated by laser synthesis are also directly applicable to bulk synthesis methods, although there are likely some differences as discussed below. It is important to note that all synthesis methods (e.g., arc discharge and laser synthesis) involve doped graphite as the starting material. Furthermore, it is the investigation of doped graphite that reveals that the C_{60} cage is completely bypassed during bottom-up self-assembly, which demonstrates how medium-size cages can form in inherently higher yield than conventional mono-metallofullerenes. These experiments are also used to disclose the cluster size effect in the initial cage nucleation step, that Sc_3N has superior nucleation properties compared to Y_3N , etc.

The title has been adjusted to more accurately reflect that doped graphite is the starting material investigated in this work to discern the origin of cluster-encapsulated fullerene cages. Further, we believe that the title expresses the major aim of the paper to a broad readership, with the details of the work clarified in the abstract.

Reviewer #2

2. The discussion about the cage structure evolution is speculative, as the structural evidence/information on the geometric framework of clusterfullerenes formed in-suit in the gas phase is insufficient. For example, the signal of $\text{Sc}_3\text{N}@c80$ found in the gas phase by laser ablation of graphite is hard to be assigned as $\text{Sc}_3\text{N}@I_h\text{-c80}$ directly. I suggest the authors should tone down to declare the cage structure of formed clusterfullerenes, or additional experiments should be added.

Author Response:

The $\text{Sc}_3\text{N@C}_{80}$ species is generated from doped graphite as a magic numbered clusterfullerene. The molecular behavior of $\text{Sc}_3\text{N@C}_{68}$, $\text{Sc}_3\text{N@I}_h\text{-C}_{80}$, and $\text{Sc}_3\text{N@D}_{5h}\text{-C}_{80}$ are explicitly probed to gain further structural insight into the $\text{Sc}_3\text{N@C}_{2n}$ formation distribution from doped graphite. The growth of $\text{Sc}_3\text{N@C}_{68}$ into larger clusterfullerenes also yields $\text{Sc}_3\text{N@C}_{80}$ as a magic numbered species. The $\text{Sc}_3\text{N@I}_h\text{-C}_{80}$ isomer is explicitly shown to be resistant to further growth into larger $\text{Sc}_3\text{N@C}_{2n}$ and thus would act as a ‘bottleneck’ in bottom-up formation routes. The $\text{Sc}_3\text{N@D}_{5h}\text{-C}_{80}$ isomer appears to be slightly more reactive. Thus, these high symmetry isomers, and $I_h\text{-C}_{80}$ in particular, are expected to be major contributing species to $\text{Sc}_3\text{N@C}_{80}$ from doped graphite. However, additional C_{80} isomers have likely formed, as the reviewer points out.

To further clarify that other Sc_3N -containing C_{80} isomers may have formed, text has been adjusted in the Abstract, Introduction, Results, and Discussion.

i) Abstract

“The conversion of doped graphite into ~~an icosahedral~~ a C_{80} cage is shown to occur through bottom-up self-assembly reactions for the first time.”

ii) Introduction

“We disclose that the formation of high symmetry clusterfullerenes occurs through a distinct bottom-up mechanism and that high-yield formation of ~~$\text{Sc}_3\text{N@I}_h\text{-C}_{80}$~~ $\text{Sc}_3\text{N@C}_{80}$ is achieved by the complete circumvention of Buckminsterfullerene, C_{60} ,²¹ in bottom-up reaction paths.”

iii) Results

“Thus, in both cases, ~~the results indicate~~ that a stable, high symmetry (~~e.g., icosahedral~~) C_{80} cage isomer is formed.”

ii) Discussion

~~“These results~~ also supports the assignment of ~~an~~ a high symmetry cage, such as $I_h\text{-C}_{80}$ cage, as a major contributing isomer for the magic numbered $\text{Sc}_3\text{N@C}_{80}$ nanocluster formed from bulk starting materials by laser plasma synthesis, ~~although other isomers should be produced to a lesser extent, as well as to that grown from the smaller $\text{Sc}_3\text{N@D}_3\text{-C}_{68}$.~~”

Finally, we would like to add that although the structural/geometrical information obtained from the in-situ gas phase formation experiments is not definitive, as pointed out by the reviewer, our proposal for the cage structure evolution is supported by DFT computations at BP86/TZP level, a methodology which has shown to make reliable predictions for relative stabilities of nitride and other clusterfullerenes [Ref 10, Popov et al. JACS 2007, 129, 11835; Rodriguez-Forteza et al. Chem. Soc. Rev. 2011, 40, 3551; Abella et al. Inorg. Chim. Acta 2017, in press, <https://doi.org/10.1016/j.ica.2017.05.040>].

Reviewer #2

3. In figure 1, the synthetic scheme for clusterfullerene is misleading. The metal and nitrogen source are simply physically mixed with graphite, whereas in this scheme it hits the intercalated graphite complex. It should be revised.

Author Response:

Fig. 1a has been revised to clarify that the starting material is doped graphite that is prepared by physical mixing. It is further emphasized in the main text, methods section, and supplementary information.

The caption of Fig. 1a has been revised as follows for clarity: “(A) Synthesis schematic for clusterfullerenes formed from a mixture of graphite, metal oxide, and melamine (nitrogen-source) in this work.”

Reviewer #3

The authors are submitting findings that address a very serious question in the field of endohedral metallofullerenes. For several decades there has been an ongoing debate regarding either a "bottom-up" or a "top-down" mechanism for their synthesis. This submission presents its findings to be consistent with a "bottom up" approach. Historically, metallofullerenes have been mostly created by either the (1) laser method [this paper] or by using the (2) electric-arc approach.

The experimental and "bottom up" findings for this work are obtained by using the laser approach, a method which may or may not relate to what happens during the arc-plasma synthesis of the electric-arc approach. That is a key question. Would these two methods produce the same bottom-up findings? To what extent are these two methods relatable with their mechanisms of formation?

On page 2, top paragraph, *.....empty cage fullerenes are suppressed, and therefore, we find an inverse formation relationship between empty fullerenes and NCFs by laser synthesis. Notably, all Sc₃N@C_{2n} (C_{2n} = 68, 70, 78, 80, 82) synthesized and isolated by means of the arc discharge methods correspond to the observed cages sizes".....*In response to this paragraph, it may or may not be relevant that an electric-arc paper (Figure 3, JACS, 129, 16257-16259, 2017) also found an inverse formation of C₆₀ and the NCF Sc₃N@C₈₀. In terms of other empty-cage fullerenes, an inverse formation of Sc₃N@C₈₀ NCF and non-C₆₀ fullerenes was observed only at very high additive levels (80%). The only reason for mentioning this reference is to convey that the laser approach and arc reactor are inherently different, but they could still be similar with regard to the bottom-up mechanism and the laser findings of this manuscript.

Overall, I find this submission to be highly meritorious in its scientific impact and intellectual contribution. The findings address a haunting question of the mechanism of endohedral metallofullerene formation. The references are well-cited, the science is sound, and the paper is of broad interest.

In summary, I support acceptance of this manuscript. For perspective, this is the most interesting manuscript that I have read in the last 10+ years. These are exciting findings.

Author Response:

We agree that there must certainly be some differences for arc discharge compared to the present laser vaporization synthesis of clusterfullerenes. As previously described, more carbon vapor may be produced, and reactions may occur in the solid state or after clusterfullerenes formed in the gas phase fall into soot (or other unknown process). Core-drilled rods packed with graphite combined with metal oxide, copper, and copper nitrate are used in the fascinating paper that the reviewer mentions. It is under the conditions of 80% copper nitrate doping that a complete inverse relationship between C₆₀ is observed, as the reviewer notes. Although we propose that the fundamental carbon chemistry and mechanistic processes that transform doped graphite into cluster encapsulated cages, such as Sc₃N@C₈₀, elucidated by the present laser-based method are applicable to arc discharge, the specific graphite-based starting materials in the aforementioned study (i.e., the CAPTEAR method) are not used in our present work. Future studies should help further compare and contrast any differences in the arc discharge and laser synthesis techniques.

To clarify that we are referring to the present experiments that use melamine as the nitrogen source, we have adjusted the text as follows:

"Under the present clusterfullerene-generating conditions, empty cage fullerenes are suppressed, ~~and therefore, we find an inverse formation relationship between empty fullerenes and NCFs by laser synthesis,~~ similar to observations for arc discharge synthesis from doped graphite containing these particular starting materials."

REVIEWERS' COMMENTS:

Reviewer #1 (Remarks to the Author):

I satisfy the revisions of this paper, and suggest to publish it as is.

Reviewer #2 (Remarks to the Author):

The authors have nicely addressed the referees' concerns and, in particular, those from my side. The title has been adjusted properly. The discussion about the cage structure was toned down in this version. In addition, the fig 1 was revised as well. Therefore, I recommend its acceptance in Nature Communications.

Reviewer #3 (Remarks to the Author):

This revised manuscript satisfies concerns that I raised in my previous review of this work. As I stated in my earlier review, this "bottom-up" assembly and experimental data is very significant in understanding endohedral metallofullerene synthesis. The work done has a nice blend of experimental and theory, and the literature citations are appropriate and helpful. The research is significant and very exciting. I look forward to seeing this manuscript published soon in Nature Communications.

Author Responses to Reviewer Comments

Reviewer #1

I satisfy the revisions of this paper, and suggest to publish it as is.

Author Response:

We gratefully thank the Reviewer for his/her comments.

Reviewer #2

The authors have nicely addressed the referees' concerns and, in particular, those from my side. The title has been adjusted properly. The discussion about the cage structure was toned down in this version. In addition, the fig 1 was revised as well. Therefore, I recommend its acceptance in Nature Communications.

Author Response:

We gratefully thank the Reviewer for his/her comments.

Reviewer #3

This revised manuscript satisfies concerns that I raised in my previous review of this work. As I stated in my earlier review, this "bottom-up" assembly and experimental data is very significant in understanding endohedral metallofullerene synthesis. The work done has a nice blend of experimental and theory, and the literature citations are appropriate and helpful. The research is significant and very exciting. I look forward to seeing this manuscript published soon in Nature Communications.

Author Response:

We gratefully thank the Reviewer for his/her comments.